# Neurotrauma clinicians' perspectives on the contextual challenges associated with long-term follow-up following traumatic brain injury in low-income and middle-income countries: a qualitative study protocol

Brandon George Smith [1,2] Charlotte Jane Whiffin [2,3] Ignatius N Esene,[4] Claire Karekezi,[5] Tom Bashford,[2] Muhammad Mukhtar Khan,[2,6] Davi Jorge Fontoura Solla,[2,7] Bhagavatula Indira Devi,[2,8] Peter John Hutchinson [1,2] Angelos G Kolias [1,2] Anthony Figaji,[2,9] Andres M Rubiano [2,10]

BGS and CJW are joint first authors.

For numbered affiliations see end of article.

**Correspondence to**
Dr Angelos G Kolias;
ak721@cam.ac.uk

## ABSTRACT

**Introduction** Traumatic brain injury (TBI) is a global public health concern; however, low/middle-income countries (LMICs) face the greatest burden. The WHO recognises the significant differences between patient outcomes following injuries in high-income countries versus those in LMICs. Outcome data are not reliably recorded in LMICs and despite improved injury surveillance data, data on disability and long-term functional outcomes remain poorly recorded. Therefore, the full picture of outcome post-TBI in LMICs is largely unknown.

**Methods and analysis** This is a cross-sectional pragmatic qualitative study using individual semistructured interviews with clinicians who have experience of neurotrauma in LMICs. The aim of this study is to understand the contextual challenges associated with long-term follow-up of patients following TBI in LMICs. For the purpose of the study, we define 'long-term' as any data collected following discharge from hospital. We aim to conduct individual semistructured interviews with 24–48 neurosurgeons, beginning February 2020. Interviews will be recorded and transcribed verbatim. A reflexive thematic analysis will be conducted supported by NVivo software.

**Ethics and dissemination** The University of Cambridge Psychology Research Ethics Committee approved this study in February 2020. Ethical issues within this study include consent, confidentiality and anonymity, and data protection. Participants will provide informed consent and their contributions will be kept confidential. Participants will be free to withdraw at any time without penalty; however, their interview data can only be withdrawn up to 1 week after data collection. Findings generated from the study will be shared with relevant stakeholders such as the World Federation of Neurosurgical Societies and disseminated in conference presentations and journal publications.

## Strengths and limitations of this study

► This study is among the first to examine the contextual challenges associated with long-term follow-up of patients following traumatic brain injury in low/middle-income countries (LMICs).

► Findings generated from this study will enable us to establish an understanding of the ways long-term follow-up of neurotrauma outcomes can be successfully recorded in LMICs, and may facilitate the ongoing development of cost-effective, novel follow-up technologies.

► Through our non-probability sampling technique and necessity for a small sample size, not all LMICs will be represented in this study, however an economic and geographical spread is sought.

► Due to the resources available and the need to conduct in-depth interviews and subsequent qualitative analysis, it was not possible to recruit non-English-speaking clinicians.

## INTRODUCTION
### Background

Traumatic brain injury (TBI) is a global public health concern; however, low/middle-income countries (LMICs) face the greatest burden, where 85% of the world's population live.[1 2] Each year, an estimated 69 million cases of TBI are reported globally,[3] with a substantial portion of these cases from falls and road traffic incidents.[3 4] In 2017, there were 521 million cases of all non-fatal injuries, a sharp increase from 354 million reported in 1990. When considering disability-adjusted life years (DALYs) from these injuries, road incidents

and falls were found to be the main contributors. Haagsma *et al* demonstrate that globally, age-standardised DALY rates increase with decreasing Sociodemographic Index.[5] The WHO recognises the significant differences between patient outcomes following injuries in high-income countries (HICs) versus those in LMICs,[6] and that almost 90% of deaths occurring from injuries occur in LMICs.[7 8] In HICs, improvements in surgical interventions and medical management mean rapid increases in survival rates and improved outcomes. In LMICs, fewer resources, different treatment practices and reduced availability of surgical interventions mean these countries have not all seen the same improvements in outcome post-TBI.[9] Despite a lack of robust data in the literature base to accurately be able to describe the incidence and mortality rates from TBI in LMICs, proportionally, LMICs experience three times as many cases of TBI when compared with HICs.[3]

Acknowledging heterogeneity within both LMICs and HICs, as a whole outcome data, is generally not reliably recorded in LMICs, and despite improved injury surveillance data, data on disability and long-term functional outcomes remain poorly recorded.[10] Therefore, the full picture of outcome post-TBI in LMICs is largely unknown. Laytin *et al* described the difficulty of collecting outcome data in resource-poor settings, specifically sub-Saharan Africa. Collection of outcome data in these settings was typically conducted within the emergency department or at hospital discharge rather than at points after discharge. Challenges of collecting outcome data included weak healthcare infrastructure and limited regular follow-up of trauma patients.[10]

Despite such challenges, the collection of long-term outcomes is essential, as it provides epidemiological data to properly determine the full burden of injury and assess the efficacy of patient treatment and management decisions in LMICs.[10 11] In addition, Mock *et al*[6] argue that to fully understand the full burden of disease, disability must be evaluated as well as death. The collection of long-term outcomes is additionally important in the implementation and continuation of registries. These resources serve to provide opportunities for continuous quality improvement and enable trials of interventions in these populations—the results of these registries can themselves be incorporated into legislation and specific care pathways.[12] Therefore, ways must be found to collect more complex long-term outcomes following TBI in LMICs that are directly comparable with the outcome data being collected in most HICs. Further adding to this challenge is a lack of clarity in the global definition of 'long-term' in neurotrauma, and what period this includes. A study of long-term outcomes after paediatric head trauma in Uganda defined long-term as anything 1–2 years post-injury,[13] while studies from France and Sweden suggest long-term as periods spanning at least 3 years and beyond.[14 15]

In turn, we must be mindful of the complex relationship between the economic status of each country and the quality of their neurosurgical follow-up provision. Many LMICs have highly heterogeneous health systems, and countries with a low gross national income per capita may still support well-functioning neurosurgical centres offering a high level of follow-up care. Although, on average, poorer countries might be expected to demonstrate worse follow-up as a function of resource limitation, we must be careful not to assume an association at the hospital level.

## Rationale

This study is the first in a three-phase project which aims to explore long-term follow-up in LMICs, its challenges, facilitators and technological solutions to augment current practices. The existing literature appears to support an association between the utilisation of trauma registries, quality improvement programmes and improved long-term outcomes. This association has been used to suggest a role for systems science approaches to improving neurotrauma care, particularly in LMICs.[8 12 16] A preliminary review of the evidence base revealed how little empirical literature had been published about the difficulties long-term follow-up poses within LMICs, and what specific contextual challenges exist within these countries. Without an in-depth understanding of these contextual challenges, it was not possible to begin work on technological solutions to improve the collection of long-term outcomes in these countries. Therefore, an exploratory qualitative study was designed to provide an in-depth understanding of neurotrauma clinicians' attitudes, experiences and perspectives of follow-up. For the purpose of this study, a 'neurotrauma physician' is defined as any medical doctor or surgeon directly involved in the provision of acute neurotrauma care. This foundation of understanding will make a meaningful contribution to the limited empirical evidence base while also informing later phases of the project examining technological solutions in low-resource settings using quantitative methods.

This project is part of the National Institute for Health Research (NIHR) Global Health Group on Neurotrauma (GHRGN) and contributes to its strategic aims. The NIHR GHRGN commenced in 2017 aiming to understand current systems and challenges in TBI clinical care within LMICs and as part of this acknowledged that the collection of long-term outcomes needed to be improved. Although beyond the scope of this particular project, results of this study may warrant an investigation into how well long-term follow-up is conducted across the spectrum of HIC neurosurgical centres, that are not directly conducting or engaging in long-term outcome studies for objective comparison. In addition, we will be careful to ensure our results are not compared with presumptions, or to centres with follow-up efforts motivated only by engagement in international studies.

## Theoretical framework

This study uses a qualitative methodology which is especially useful to explore complex phenomena[17] when little is known about a topic or where new insights are required. Qualitative inquiry is committed to subjectivity and the collection of rich in-depth data through the examination

of participant's viewpoints.[17] Pragmatic qualitative research, also known as descriptive and generic qualitative research, does not align itself to any specific philosophical perspective and therefore allows flexibility of methods that enables the study to be designed in an appropriate way to address the research question.[18] Pragmatic research lies within the naturalistic paradigm which recognises the existence of multiple realities and studies real-world situations by having little impact on the phenomena under investigation.[19] Pragmatic qualitative studies aim to achieve in-depth understanding prioritising literal description first and then understanding phenomena in a deeper sense through analysis and interpretation of how people draw meaning from their experiences.[20] The goal of such research is to provide an account of the 'experiences, events and process that most people (researchers and participants) would agree are accurate' (p128).[21] In addition, Sullivan-Bolyai *et al*[21] state qualitative descriptive designs are particularly useful when researching populations of other cultures and as such would be well suited to the aims of this study.

### Research aim

The aim of this study is to understand the contextual challenges associated with long-term follow-up of patients following TBI in LMICs. For the purpose of the study, we define 'long-term' as any data collected following discharge from hospital.

### Objectives

1. To determine current approaches to long-term follow-up of patients following TBI in LMICs.
2. To explore the challenges associated with long-term follow-up of patients following TBI in LMICs.
3. To collaboratively offer recommendations for appropriate solutions that will facilitate long-term follow-up of patients following TBI in LMICs based on findings and themes identified.

### METHODS AND ANALYSIS
### Study design

We propose a cross-sectional pragmatic qualitative study using individual semistructured interviews with clinicians who have experience of neurotrauma in LMICs. Despite the pragmatic nature of this study, the naturalistic paradigm in which it is situated aims to interpret individual experiences and employs a subjective epistemology which asserts universal knowledge is not possible.[22] A purposive sampling strategy will be employed to recruit 24–48 participants from LMICs participating in the NIHR GHRGN. Methods include semistructured telephone/online video interviews and the collection of relevant demographic and professional data.

### Data collection

This study will collect data through semistructured telephone or online video interviews, conducted by a single interviewer (BGS). Semistructured interviews are the cornerstone of qualitative methodology as they facilitate rich data

---

> **Box 1  Demographic data**
>
> ► Age.
> ► Sex.
> ► Country of residence.
> ► Occupation.
> ► Years of experience.
> ► Years working in neurotrauma.
> ► Practice setting.

gathering and in-depth exploration of the phenomenon under investigation. As such semistructured interviews were the natural choice for data collection in this study. Telephone and online video methods were chosen over face-to-face for pragmatic reasons given the geographical location of the participants and the interviewer. We will also collect some limited demographic data that will help us analyse and interpret the similarities and differences between the data sets (see box 1).

Specific demographic data will be collected using an electronic form (see box 1).

Details of the interview questions are presented in box 2, and participants will be made aware of these prior to the interview so they have the opportunity to consider their response. Questions are open ended to encourage a rich dialogue and participants will be free to elaborate on their answers adding additional detail as required. To encourage more in-depth responses, prompts, probes and follow-up questions will be used. In this sense the questions in box 2 will be used more as a guide than an explicit framework, which is consistent with a semistructured approach. It is anticipated that the interviews will last up to 60 min and will be conducted by BGS following a pilot of the interview guide. Interviews will be recorded using a digital recorder.

### Study setting

Participants will be recruited from any country identified as low or middle income as defined by the 2017–2018 World Bank list of economies.[23] Access to participants will be negotiated using the research group's project lead and collaborators from the 12 countries in the NIHR GHRGN listed in box 3. New collaborators in the group, including the Philippines and Zimbabwe, in addition to participants from institutions from LMICs not yet part of the group, will be invited to participate as necessary in order to attain the defined sample size.

### Eligibility criteria

To be eligible for this study, the person must be a practising physician within an LMIC and have experience of managing neurotrauma, so they have experience of treatment and discharge of patients with TBI. We are requesting the clinicians all have at least 2 years' experience of neurotrauma and also some experience of research, so they have more insight into the complexities of long-term follow-up of neurotrauma patients and understand the research process. Due to availability of resources, participants will have to be

## Box 2  Interview guide

- ► Current follow-up.
- A. Please can you tell me about how and when your patients are currently followed-up following discharge from a traumatic brain injury (TBI)?
  - a. What outcomes are measured?
  - b. What needs are assessed?
  - c. What referrals are made?
  - d. What type of contact system do you have in place (face-to-face, telephone, telemedicine)?
- ► Definition and understanding of long-term outcomes.
- A. What do you understand by the use of 'long-term' in the context of outcomes post-TBI?
- B. Do you think your definition and understanding of 'long-term outcomes' in your country are the same as those in other countries?
- C. Do you know or use any type(s) of outcome classification for follow-up?
- ► Attitudes towards long-term follow-up.
- A. What do you think about the need for long-term follow-up of patients following TBI in your setting?
- B. What would you consider to be the main benefits of long-term follow-up of patients with TBI?
- C. What do you consider to be the challenges associated with long-term follow-up of your patients with TBI?
- D. Are these challenges more related to: health system aspects, national, regional, or local administrative aspects, institutional aspects, resources for care aspects?
- E. Are there any other issues that need to be considered for long-term follow-up in your own setting and wider country?
- ► Long-term follow-up in low/middle-income countries (LMICs).
- A. Thinking more broadly now about other LMICs, do you have any other thoughts about long-term follow-up in these other LMICs?
- B. What else might be required in LMICs to facilitate long-term follow-up of patients following TBI?
- ► Possible interventions to facilitate follow-up in LMICs.
- A. Do you know what types of technology patients and families have access to that could help in follow-up of patients following TBI?
- B. What kind of technology would help in your country to record long-term follow-up?
- C. Are you aware of any specific mobile health, telemedicine or phone interview options for long-term follow-up in your institution/state/country?
- D. Does your service or institution participate in a trauma or neurotrauma quality improvement programme including the use of clinical registries with outcome measures?

## Box 3  Low/middle-income countries in the National Institute for Health Research Global Health Research Group on Neurotrauma

1. Brazil
2. Colombia
3. Ethiopia
4. India
5. Indonesia
6. Malaysia
7. Myanmar
8. Nigeria
9. Pakistan
10. South Africa
11. Tanzania
12. Zambia

Sullivan-Bolyai *et al*[21] recommend a sample size of 20–50 participants.

Moser and Korstjens[24] explain that a guiding principle of sampling in qualitative studies is to sample up until data saturation is reached. Data saturation is a qualitative principle which is a point whereby the researchers are confident that new data will not reveal new information.[25] However, data saturation is a contentious issue in reflexive thematic analysis (TA) which advocates a less prescriptive approach to sampling and asks the researcher to make an interpretive judgement about when to stop.[26]

In line with this view, sample size in this study was further informed by a desire to explore a variety of views from clinicians within a range of LMICs. A non-probability, purposive sample is therefore required. We will aim to have representation from a range of countries and, if possible, a range of regions within these countries. Therefore, we propose a sample size of 24–48 participants and will comment further on principles of data saturation in the final paper.

### Recruitment

We will begin by seeking to recruit two to three neurotrauma clinicians from the collaborating countries in the NIHR GHRGN, consisting of 12 LMICs (box 3), with new collaborators from Zimbabwe and the Philippines. We will also be placing a call through social media (Twitter and WhatsApp) to maximise the opportunity for recruitment from any clinicians not subscribed to the GHRGN mailing list or from other LMICs not listed as collaborating regions.

The local primary investigators (PIs) will be written to through the group's previously established mailing lists, with information pertaining to the study including the aims of the study, inclusion criteria (box 4) for participants and data collection methods. We will then ask the local PIs to forward information about the study to relevant clinicians (NIHR GHRGN collaborators are also eligible to participate). For social media recruitment, an infographic was developed for the study and distributed through Twitter as an open call for participants. Those who wished to express an interest can contact the

fluent in English. Experience of working with collaborators would suggest this criterion will not limit participation in the study. All participants will have to provide written informed consent by agreeing to the criteria within the electronic consent form.

### Sample

The main criticism of qualitative studies stems from the necessity for small samples. Smaller samples facilitate depth of inquiry and rich interpretation. However, according to Sullivan-Bolyai *et al*,[21] qualitative descriptive studies can accommodate more moderate sample sizes than theoretically informed studies because there is no requirement to generate theory. Therefore,

**Box 4    Inclusion criteria**

► Physician within a low/middle-income country with experience of managing neurotrauma.
► At least 2 years' experience of neurotrauma.
► Experience of collecting, or attempting to collect, traumatic brain injury outcome data.
► Self-declared fluency in spoken English.
► Able to provide informed consent.

**Box 5    Braun and Clarke's[27] thematic analysis framework**

1. Familiarising yourself with your data.
2. Generating initial codes.
3. Searching for themes.
4. Reviewing themes.
5. Defining and naming themes.
6. Producing the report.

lead author (BGS) for more information. Information forwarded to participants will include a letter of invitation, participant information sheet and contact details for the research team. If interested in participating, an initial telephone call will be arranged to review the participant information sheet and the requirements of the study. If the physician still wishes to continue, we will schedule a telephone interview. We will leave a minimum of 1 week in between this initial contact and arrangement of the interview to allow a cooling-off period. If an insufficient number of people have responded to the first call for participants, a second request for participation will be made 2–3 weeks later. Data will be collected over a 12-month period, beginning March 2020.

It is essential that participants join this study of their own free will. However, the notion that junior colleagues may have a 'reciprocal obligation' to assist their more senior colleagues, and the institution, in participating in global research is appreciated. To safeguard against potential coercion, we will highlight to our participants that they are under no obligation to take part. We will discuss this specifically during the preconsent meeting and ensure they understand that no one will be informed if they chose, or do not choose, to participate in the study. Participants will not be provided with any financial reimbursement, as no out-of-pocket expenses are anticipated.

### Data analysis
Analysis within pragmatic qualitative studies should stay close to the text, and does not require 'highly abstract rendering of data'.[20] Staying close to the text means early coding decisions rely heavily on what participants actually said with interpretation applied more substantially in later phases of analysis. Studies which are independent of theory can and often do use TA.[17 27] According to Braun and Clarke,[27] TA can reveal a rich, detailed and complex exploration of the data and despite its 'descriptive' title will end with some necessary interpretation of the data[20] through discovery of shared and varied patterns of understanding.[28] Indeed, Vaismoradi *et al*[17] contend that thematic approaches actually attend little to the description of the data set and instead require interpretation of the various aspects of the phenomena under investigation. It is therefore proposed that a six-stage Braun and Clarke[27] TA will be conducted (see box 5). Congruent with Braun and Clarke's more recent commentary on

TA, this study aligns itself to an approach described as 'reflexive TA'.[29]

To complete the analysis, audio files from interviews will first be transcribed verbatim by a transcription service and checked for accuracy by BGS. Codes will be assigned to individual transcripts and will be supported by the use of NVivo software, allowing researchers to organise the data, share coding decisions and confirm the origins of interpretation. Similar codes will be grouped together, and initial themes identified. Themes will then be reviewed, revised and final themes named and agreed on within the research team. BGS will take lead on the data analysis, supported by CJW in the initial coding stages. This process of coding and theme building is not about reaching consensus between coders about the way codes were applied or their interpretation.[29] Instead, this is a collaborative process to advance understanding of the data and develop a more nuanced and reflexive interpretation congruent with our epistemological position. We will then report our final interpretive themes and the supporting subthemes identified in the analysis. These will be defended through critical discussion and the origins of the themes evidenced through the presentation of anonymised direct quotes. A more in-depth discussion of the data analysis process and audit trail will be presented in the findings of the paper.

BGS is a PhD student and fourth year medical student who has undertaken training in qualitative methods and analysis. CJW is an experienced qualitative researcher and nurse academic.

### Rigour
Validity, generalisability and reliability have little relevance in qualitative research.[30 31] Instead qualitative research is judged by more appropriate criteria such as credibility and trustworthiness.[32] These hallmarks of quality determine the rigour of a qualitative study. In this study, we intend to include a number of strategies advocated by Nicholls[32] to increase the rigour of the methods and consequently the final findings. First, following a reflexive, interpretive approach our findings will be strengthened through the critical dialogue between members of the research team, principally BGS and CJW, and subsequently with our coauthors. In addition, we propose using member checking as advocated by Lincoln *et al*[33] as crucial to establishing credibility in qualitative studies and according to Birt *et al*[34] can overcome issues of researcher bias and increase transferability of findings. In this regard, we will first return any transcripts where transcription was not possible due to

the quality of recording to ask for clarification of specific sections of the interview if necessary. Second, we will share initial themes identified from the data with participants so they can add further insight to the interpretation of findings if they wish. These contributions will be added to the final analysis of data.

## ETHICS AND DISSEMINATION

This is an open, online study with global participants, with the PI and principal research team all based in the UK. Although including overseas respondents, as this research is hosted and delivered by the UK, favourable opinion is only sought from a UK ethics committee.

Subsequently, in February 2020, the University of Cambridge Psychology Research Ethics Committee reviewed this study and provided favourable ethical opinion (Ref: PRE.2020.010), and thus is in line with the Concordat to Support Research Integrity from Universities UK, Research Councils UK and the UK Research Integrity Office (2013), International Ethical Guidelines for Health-related Research Involving Humans, Council for International Organizations of Medical Sciences 2016, the World Medical Association's Declaration of Helsinki and Good Clinical Practice (the ICH GCP R2 2016) as applicable. The University of Cambridge is the sponsor and appropriate insurance is in place. Ethical issues within this study include consent, confidentiality and anonymity, and data protection.

All participants will be required to provide written informed consent after a cooling-off period of at least a week after the participant information sheet has been delivered. Participants can withdraw at any time without penalty, with a grace period of a week given after completion of the interview to withdraw their interview data, after which withdrawal of data will not be possible as analysis will have commenced.

All information will be kept strictly confidential, with all information generated to be stored on a General Data Protection Regulation- compliant confidential server. A unique study ID will be allocated to all participants and used to anonymise personal data. Audio files will be immediately transferred to an encrypted folder and the data on the recording device destroyed, interview data will be anonymised upon transcription and other personal information changed or removed once checked by the research team. Participants will be told that anonymised quotes will be published in the findings of this study; however, despite this anonymity it is possible that their contributions may be recognisable to others reading the study. We will also be seeking consent to disclose region and level of income associated with their country of origin against participant quotes used in the publication of findings, as this will be important to the contextual understanding of the study.

### Patient and public involvement

This study seeks to ascertain the views of clinicians, and thus patient and public involvement was not considered to be as relevant in this context. However, we recognise the value of working with key stakeholders to develop research and therefore asked for peer review of this study by collaborating members of the GHRGN. Their comments informed the final study design.

### Dissemination

When we have completed the study, we will produce a study summary which we will offer a copy to participants via email. Study findings will also be disseminated internationally through the GHRGN network and other relevant stakeholders such as the World Federation of Neurosurgical Societies, in addition to appropriate conferences, peer-reviewed journal publications and social media.

### Study limitations

Understandably, through the non-probability approach to sampling and the necessity for a small sample size, not all LMICs will be represented in this study, however an economic and geographical spread is sought. We anticipate developing considerable insight from our international cohort of participants with differing levels of income and localities, in addition to establishing an understanding of the ways long-term follow-up of neurotrauma outcomes can be successfully recorded in LMICs. Careful attention to nuances must be paid. We respect and have the foresight that there will be heterogeneity within LMICs as a whole, and even within countries themselves as to the current provision, capabilities and attitudes towards neurotrauma follow-up. We must be sensitive when categorising our findings across a spectrum based on the gross national income per capita. We acknowledge that some false dichotomy may exist in the dialogue regarding healthcare provision, particularly of the collection of long-term outcomes, and caution will be exercised in interpretation of our findings. Unfortunately, due to the resources available and the need to conduct in-depth interviews and subsequent qualitative analysis, it is not possible to recruit non-English-speaking clinicians.

Furthermore, we recognise that this study may present a narrow perspective on long-term follow-up by only presenting the views of neurotrauma clinicians. Given the dearth of literature in this field, this study will still make a valuable contribution to the evidence base. However, future studies are encouraged that are inclusive of professional groups such as nurses, allied health professionals and data managers/epidemiologists who may have a wider view on neurotrauma follow-up in LMICs.

Notwithstanding, to our knowledge this study is among the first to examine the contextual challenges associated with long-term follow-up of patients following TBI in LMICs. Findings generated from this study will enable us to establish an understanding of the ways long-term follow-up of neurotrauma outcomes can be successfully recorded in LMICs. This knowledge may lend itself to facilitating the ongoing development of cost-effective and novel follow-up technologies in the future.

**Author affiliations**
[1]Department of Clinical Neurosciences, Division of Neurosurgery, Addenbrooke's Hospital, Cambridge, UK

[2]NIHR Global Health Research Group on Neurotrauma, University of Cambridge, Cambridge, UK

[3]College of Health of Social Care, University of Derby, Derby, UK

[4]Neurosurgery Division, Faculty of Health Sciences, University of Bamenda, Bambili, Northwest Region, Cameroon

[5]Department of Neurosurgery, Rwanda Military Hospital, Kigali, Rwanda

[6]Department of Neurosurgery, Northwest General Hospital and Research Center, Peshawar, Pakistan

[7]Department of Neurosciences and Behaviour Sciences, University of São Paulo, Ribeirao Preto, Brazil

[8]Department of Neurosurgery, National Institute of Mental Health and Neuro Sciences, Bangalore, India

[9]Division of Neurosurgery, University of Cape Town, Rondebosch, South Africa

[10]Neurosciences Institute, Department of Neurosurgery, El Bosque University, Bogota, Colombia

**Contributors** The concept of this study was conceived by PJH, AGK, AF and AMR as part of Theme 3D of the NIHR Global Health Research Group on Neurotrauma, with further input by BGS, CJW, INE, CK, TB, MMK, DJFS, BID. CJW wrote the first draft of the protocol, with BGS preparing the protocol for publication. All authors approved the final manuscript.

**Funding** This research was supported by the National Institute for Health Research (NIHR) Global Health Research Group on Neurotrauma (grant number 16/137/105) using UK aid from the UK government.

**Disclaimer** The views expressed in this manuscript are those of the authors and are not necessarily those of the UK National Health Service, NIHR or the Department of Health.

**Competing interests** AGK and PJH are supported by the National Institute for Health Research (NIHR) Cambridge Biomedical Research Centre and the NIHR Global Health Research Group on Neurotrauma. PJH is also supported by an NIHR Research Professorship. The NIHR Global Health Research Group on Neurotrauma was commissioned by the UK NIHR using Official Development Assistance funding (project no. 16/137/105). INE, CK, MMK, DJFS and AGK are members of the Young Neurosurgeons Committee of the World Federation of Neurosurgical Societies. The committee is supporting this project.

**Patient and public involvement** Patients and/or the public were not involved in the design, or conduct, or reporting, or dissemination plans of this research.

**Patient consent for publication** Not required.

**Provenance and peer review** Not commissioned; externally peer reviewed.

**Open access** This is an open access article distributed in accordance with the Creative Commons Attribution 4.0 Unported (CC BY 4.0) license, which permits others to copy, redistribute, remix, transform and build upon this work for any purpose, provided the original work is properly cited, a link to the licence is given, and indication of whether changes were made. See: https://creativecommons.org/licenses/by/4.0/.

**ORCID iDs**
Brandon George Smith http://orcid.org/0000-0001-8471-1368
Charlotte Jane Whiffin http://orcid.org/0000-0002-9767-2123
Peter John Hutchinson http://orcid.org/0000-0002-2796-1835
Angelos G Kolias http://orcid.org/0000-0003-3992-0587
Andres M Rubiano http://orcid.org/0000-0001-8931-3254

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
