## [Reviewer comments · BMJ Open]

ARTICLE DETAILS

TITLE (PROVISIONAL)	Neurotrauma physicians' perspectives on the contextual challenges associated with Long-term follow-up following Traumatic Brain Injury in Low- and Middle-income countries: a qualitative study protocol
AUTHORS	Smith, Brandon; Whiffin, Charlotte; Esene, Ignatius; Karekezi, Claire; Bashford, Tom; Mukhtar Khan, Muhammad; Fontoura Solla, Davi; Indira Devi, Bhagavatula; Hutchinson, Peter; Kolia, Angelos; Figaji, Anthony; Rubiano, Andres M

VERSION 1 – REVIEW

REVIEWER	John Yue University of California San Francisco, USA
REVIEW RETURNED	24-Jun-2020

GENERAL COMMENTS	Interesting study on neurotrauma physicians' understanding of available resources and perceptions toward long-term follow-up of TBI patients in their country. The research question is important. Several elements remain to be addressed: In the introduction the authors state "Outcome data is not reliably recorded in LMICs and despite improved injury surveillance data, data on disability and long-term functional outcomes remain poorly recorded. Therefore the full picture of outcome post-TBI in LMICs is largely unknown." I'm not certain that the current list of questions addresses this shortcoming stated in the introduction. The current questions focus mostly on physician attitudes rather than detailed and specific assessment of the strengths and weaknesses of the current resource allocation and framework for follow-up in the respective country. I think the questions should include quantitative numbers as well as a list of regional, institutional, and financial limitations to follow-up. The authors state they plan to interview 24-48 physicians, however the distribution of physicians and countries are unknown. The authors should have a concrete framework of how they plan to reach out to each LMIC country and seek to capture data from each of the listed LMIC countries in their table. What do the authors plan to do with the qualitative data? Is a qualitative summary planned to serve as the results for this study?
---

REVIEWER	Shivanthi Balalla Auckland University of Technology, New Zealand
REVIEW RETURNED	19-Jul-2020

GENERAL COMMENTS	Reviewer comments: This protocol focuses on highlighting the challenges of long-term follow-up care in TBI outcomes in LMICs. It is agreed that evidence from these countries is sparse in the literature and therefore the current protocol aims to address an important gap in information. Generally, the concept of the proposed study is well conceptualised, with clear objectives and theoretical framework. In some sections there was some information missing for the reader, which are detailed below. Some suggestions are also made to break up particularly lengthy sentences. Line 9-12, Background: While you have highlighted some apparent differences between the LMICs and HICs in terms of outcomes, some more detail with some numbers on mortality rates, and outcomes would really drive this point home. Line 30-34: “In turn, we must be sensitive to the economic capacities of each country versus their quality of follow-up provision, and acknowledge that the distinction between countries by economic brackets is somewhat artificial; that the gross national income per capita does not necessarily follow the efforts and successes being made by neurosurgical departments, so as to not further encourage any existing false dichotomies.” This is a very long sentence and affects its clarity. Please try to break it down to two or three sentences. Line 40-43: “There is a growing notion in literature that a synergy exists between the utilisation of trauma registries, quality improvement programmes and long-term outcomes, and have all been advocated into systems science approaches for improving neurotrauma care in all regions, especially in LMICs where such registries and schemes are largely absent [6-8].” This sentence needs to be revised as it is lacking clarity. As above, I would suggest to break the sentence into two clear sentences. Study design There is scant information about the study design under this section. Although it is noted that this information is described in the forthcoming sections of the protocol, please provide an overview about the sampling frame (e.g. the LMIC regions considered), recruitment strategies, sampling units (trauma surgeons, or other clinical/non-clinical specialists?) and anticipated sample size. Please also state the epistemological framework guiding this research. Data collection. Please state how ethical consent will be obtained from participants and whether the ethics consent will be sought from the relevant local ethics committees . Under demographic data, I would suggest using the term “sex” instead of “gender”. Similarly with “job title”, my suggestion would be to use “occupation”, as the former is more focused on titles as opposed to giving information regarding the actual area of work/specialisation. Please also state how many interviewers will be conducting interviews and whether the same members will be involved in transcription.
--

	Rigour How will disagreements during coding be addressed? Please also include a statement of how coding agreement between the researchers is ascertained (inter-rater reliability) under this section. Eligibility criteria It is unclear why only physicians' perspectives are sought in this study. In this sense it gives a very narrow perspective of the challenges of long-term follow-up focussed purely on treatment. There does not seem to be a valid justification as to why other health professionals such as nurses, psychologists and data managers/epidemiologists (who work routinely with follow-up data) are not included in the inclusion criteria. If the focus of the study is indeed on physicians' perspectives only, then appropriate justification needs to be made clear in the rationale section, and also reflected in the title of the study. Furthermore, this limitation needs to be acknowledged under "Study Limitations". Recruitment Please state if participants are provided with any financial reimbursement.
--	--

REVIEWER	Marina Downing Monash University, Australia
REVIEW RETURNED	26-Aug-2020

GENERAL COMMENTS	I encourage the authors to read Braun & Clarke's more recent articles that cover their reflexive thematic approach, and their views on data saturation.
---

REVIEWER	Christel McMullan University of Birmingham, United Kingdom Qualitative Research Traumatic Brain Injury Patient Reported Outcomes Randomised Controlled Trial IBD
REVIEW RETURNED	06-Oct-2020

GENERAL COMMENTS	The authors need to specify how long the study will be, how long they will have to conduct data collection. I think the manuscript would benefit from a more detailed definition of 'long-term' The justification for using qualitative research, telephone/video interviews is a little weak in my opinion. I also would rephrase the sentence around 'Although face to face interviews are more common..' as it is not a justification to use face to face interviews, even if it was possible for them to do so. Box 2 - is it a summary or is it the full interview schedule? If it is the full schedule, don't say it's a summary in the text. Some of the questions are too long and would benefit from being divided into two questions or rephrased.
--

	More probes might need to be included - cultural issues for instance? Explain what 'close to the text' means (Analysis) Themes arising/emerging - please rephrase I think the sample/recruitment/eligibility criteria sections should be placed before analysis. The protocol would also benefit from more detail on how the authors will ensure that data saturation is reached?
--	--

VERSION 1 – AUTHOR RESPONSE

	Action requested	Comments	Actions	Page
Reviewer 1	Interesting study on neurotrauma physicians' understanding of available resources and perceptions toward long-term follow-up of TBI patients in their country. The research question is important.	Thank you for your kind comments.	No action required.	
	Several elements remain to be addressed: In the introduction the authors state "Outcome data is not reliably recorded in LMICs and despite improved injury surveillance data, data on disability and long-term functional outcomes remain poorly recorded. Therefore the full picture of outcome post-TBI in LMICs is largely unknown." I'm not certain that the current list of questions addresses this shortcoming stated in the introduction. The current questions focus mostly on physician attitudes rather than detailed and specific assessment of the strengths and weaknesses of the current resource allocation and framework for follow-up in the respective country. I think the questions should include quantitative numbers as well as a list of regional, institutional, and financial limitations to follow-up.	Thank you for your important observation. This study is the exploratory phase of a project which will include quantitative inquiry in phases two and three. Methodologically it is not possible to include quantitative data in this first phase. However, these findings will inform our future quantitative work. We have now described the phases of this project to ensure readers have a clearer understanding of the exploratory nature of this study and the wider work in which we are engaged.	Clarification of the purpose of the qualitative study has been added to Rationale.	Pg. 3 Pg. 4

	The authors state they plan to interview 24-48 physicians, however the distribution of physicians and countries are unknown. The authors should have a concrete framework of how they plan to reach out to each LMIC country and seek to capture data from each of the listed LMIC countries in their table.	We have made our recruitment strategy clearer by adding in additional details including our intention to select a specific number of participants from each of the collaborating countries. Although we have collaborators assisting in the recruitment of participants it is imperative that participants join of their own free will and do not feel coerced in anyway. Therefore, we are unable to further strengthen this particular approach to recruitment. However, we were able to add a further method of recruitment through submission of an amendment of the protocol to the ethics committee. This was necessary as recruitment during the COVID-19 pandemic has been particularly challenging. Therefore, we have added the new social media call for participants used in this study which will give us more opportunity to reach a wider community of physicians.	Recruitment section has been amended to provide further detail on strategy.	Pg. 7
	What do the authors plan to do with the qualitative data? Is a qualitative summary planned to serve as the results for this study?	Thank you for requesting clarity on this point. We have adjusted the manuscript to make it clear that we will be engaged in a reflexive thematic analysis, as is congruent with the later work of Braun and Clarke (2019) The expected outcome of this approach is 'themes'. We will present the themes and sub-themes which are present within these. Each will be explored in depth using anonymised direct quotes as evidence of their origin.	We have amended the Data Analysis section to include more information regarding our specific approach to data analysis and how the findings will be presented.	Pg. 7
Reviewer 2	This protocol focuses on highlighting the challenges of long-term follow-up care in TBI outcomes in LMICs. It is agreed that evidence from these countries is sparse in the literature and therefore the current protocol aims to address an important gap in information. Generally, the concept of the proposed study is well conceptualised, with clear objectives and theoretical framework.	Thank you for your kind comments.	No action required.	

	In some sections there was some information missing for the reader, which are detailed below. Some suggestions are also made to break up particularly lengthy sentences. Line 9-12, Background: While you have highlighted some apparent differences between the LMICs and HICs in terms of outcomes, some more detail with some numbers on mortality rates, and outcomes would really drive this point home.	Thank you for this very helpful suggestion, the background to the study is strengthened with the inclusion of this data.	We have added additional figures to our background.	Pg. 3
	Line 30-34: "In turn, we must be sensitive to the economic capacities of each country versus their quality of follow-up provision, and acknowledge that the distinction between countries by economic brackets is somewhat artificial; that the gross national income per capita does not necessarily follow the efforts and successes being made by neurosurgical departments, so as to not further encourage any existing false dichotomies." This is a very long sentence and affects its clarity. Please try to break it down to two or three sentences.	Thank you for identifying this point.	We have broken down this paragraph further to improve its clarity.	Pg. 3
	Line 40-43: "There is a growing notion in literature that a synergy exists between the utilisation of trauma registries, quality improvement programmes and long-term outcomes, and have all been advocated into systems science approaches for improving neurotrauma care in all regions, especially in LMICs where such registries and schemes are largely absent [6-8]." This sentence needs to be revised as it is lacking clarity. As above, I would suggest to break the sentence into two clear sentences.	Thank you for identifying this point.	We have broken down this paragraph and slightly altered its wording further to improve its clarity.	Pg. 3

	Study design There is scant information about the study design under this section. Although it is noted that this information is described in the forthcoming sections of the protocol, please provide an overview about the sampling frame (e.g. the LMIC regions considered), recruitment strategies, sampling units (trauma surgeons, or other clinical/non-clinical specialists?) and anticipated sample size. Please also state the epistemological framework guiding this research.	Thank you for identifying the limited information under this title.	We have now rectified this omission and added summary information about the type of study employed, methods involved and epistemological framework underpinning this study.	Pg. 4
	Data collection. Please state how ethical consent will be obtained from participants and whether the ethics consent will be sought from the relevant local ethics committees.	This is an open, online study with global participants, with the primary investigator and principal research team all based in the UK. Although respondents will exclusively be located overseas, as this research is hosted and delivered by the UK, only UK ethics were sought.	We have made it clear in the manuscript why only UK ethics opinion was sought and strengthened the ethical scrutiny applied.	Pg. 8
	Under demographic data, I would suggest using the term "sex" instead of "gender". Similarly with "job title", my suggestion would be to use "occupation", as the former is more focused on titles as opposed to giving information regarding the actual area of work/specialisation.	Thank you for your comment.	Amendment made please see Box One: Demographic data.	Pg. 5
	Please also state how many interviewers will be conducting interviews and whether the same members will be involved in transcription.	All interviews will be conducted by the lead author BS. We have added this to the manuscript. Transcription will be undertaken by a transcription service (Pg. 7). We have now clarified that BS will check these for accuracy before analysis commences.	Details of whom will conduct the interviews (Brandon Smith, an MBPhD student with training in qualitative methods) have been added to Data collection.	Pg. 7

	Rigour How will disagreements during coding be addressed? Please also include a statement of how coding agreement between the researchers is ascertained (inter-rater reliability) under this section.	In the later work of Braun and Clarke they clarify the expectant requirements of coding within their methodology. They state that “If more than one researcher is involved in the analytic process, the coding approach is collaborative and reflexive, designed to develop a richer more nuanced reading of the data, rather than seeking a consensus on meaning”. Therefore, following a reflexive, interpretive approach, the aim of involving a second coder will not be to check for agreement per se, but to debate, refine and advance our understanding of the raw data. Differences in the tentative codes assigned to the data by the lead and secondary coder are fundamental to the reflexive nature of the analysis. We have alluded to this point in this manuscript. However, a more in-depth discussion of this point may be more well-suited to a findings report, which will describe in detail the particular process carried out.	We have added further detail regarding involvement of a secondary coder from the research team and their role in our reflexive interpretive approach under data analysis and rigour sections of the manuscript.	Pg. 8
	Eligibility criteria It is unclear why only physicians’ perspectives are sought in this study. In this sense it gives a very narrow perspective of the challenges of long-term follow-up focussed purely on treatment. There does not seem to be a valid justification as to why other health professionals such as nurses, psychologists and data managers/epidemiologists (who work routinely with follow-up data) are not included in the inclusion criteria. If the focus of the study is indeed on physicians’ perspectives only, then appropriate justification needs to be made clear in the rationale section, and also reflected in the title of the study. Furthermore, this limitation needs to be acknowledged under “Study Limitations”.	This is a very interesting point and one that is important in this field. The decision to recruit physicians was led by the needs of the wider research works in which this study is nested. Therefore, in this study it is right to only focus on physicians. Whilst we appreciate physicians might not be the only ones involved in follow-up, they are leading care decisions and thus would have the best insight into the overall process of Long term follow up. Furthermore, as BS is a physician, he is better placed to interview other physicians as they are more likely to have open conversations without the issue of power/hierarchy which is very significant in many LMICs. We recognise the narrow perspective this has led to and have recommended future research with a wider population of professionals.	We have added further detail to our study rationale detailing the scope of the work. We have added to the limitations as suggested that whilst part of a wider series of research, this particular study focusses on physicians and recommends future studies with a wider professional group. We have modified our full title to reflect our focus on neurotrauma physician perspectives.	Pg. 1, Pg. 3, Pg. 4, Pg. 9
	Recruitment Please state if participants are provided with any financial reimbursement.	Thank you for your comment – participants will not be provided with any financial reimbursement.	We have added a statement on reimbursement under the Recruitment section.	Pg. 7

Reviewer 3	I encourage the authors to read Braun & Clarke's more recent articles that cover their reflexive thematic approach, and their views on data saturation.	Thank you for this recommendation, we have spent time reviewing Braun & Clarke's more recent papers as suggested, added further detail to the manuscript and cited key papers based on this exploration. With regard to data saturation, we were cautious in stating a proposed sample size of 24-48 participants given the stance of predisposing sample sizes in the field of qualitative research. However, we were required to provide such a range for ethical opinion. Therefore before data collection began, and thus before what we would know what the analysis of said transcripts would entail, we based our estimate on previous published literature using similar sample sizes in a similar context in addition to our ambition to represent the views of a number of different LMIC countries. At this stage we are limited in our ability to discuss this issue further. However, in our findings paper we will examine the concept of data saturation in more detail and defend why we are confident of any conclusion reached within the sample achieved.	We have alluded to the current debate surrounding data saturation and provided further justification for the sample size determined at the outset of this study.	Pg. 6
------------	--	--	--	--------------

Reviewer 4	The authors need to specify how long the study will be, how long they will have to conduct data collection.	Data will be collected over a 12-month period, beginning March 2020.	We have added clarity under 'Recruitment'.	Pg. 7
	I think the manuscript would benefit from a more detailed definition of 'long-term'	Thank you for your comment we appreciate the benefit of this definition at the start of this paper. Given the different definitions of 'long term' we purposefully avoided aligning our definition of 'long-term' with this literature as we believed this may constrain the exploratory nature of the qualitative inquiry. Instead we defined long term as any data collected following discharge from hospital (see aim p.4). We were then able to explore this definition with our participants to gain their views on what period they perceived to be 'long-term' and this will be reported on in the findings.	We have justified the lack of definition in the manuscript.	Pg. 3
	The justification for using qualitative research, telephone/video interviews is a little weak in my opinion. I also would rephrase the sentence around 'Although face to face interviews are more common..' as it is not a justification to use face to face interviews, even if it was possible for them to do so.	Thank you for asking us to strengthen the defence of our qualitative methods. We hope that further detail on the overall programme of work in which this study is nested will now address this primary concern. In terms of data gathering we have explained that the video/telephone methods were chosen for pragmatic reasons due to geographical local of the research team and participants.	Further justification given for qualitative methodology and methods.	Pg. 3 Pg. 4
	Box 2 - is it a summary or is it the full interview schedule? If it is the full schedule, don't say it's a summary in the text. Some of the questions are too long and would benefit from being divided into two questions or rephrased. More probes might need to be included - cultural issues for instance?	Thank you for asking for clarity on this point and suggesting additional probes. The interview guide presented in Box Two was put forward for ethical opinion and provided a guide for the semi-structured interview. However, as is expected in this style of interviewing a degree of flexibility is employed, questions will be re-phrased, probes and follow-up questions asked. In this sense the questions provide a guide rather than a framework or 'schedule' as you allude to in your comment.	We have removed the word summary and schedule and alluded to the more flexible data gathering process including prompts, probes and follow-up questions which are more commensurate with the style of semi-structured interviewing.	Pg. 5

	Explain what 'close to the text' means (Analysis) Themes arising/emerging - please rephrase	Thank you for requesting clarity in this point. We are using this phrasing to explain to readers that our first stage of coding relied heavily on descriptive codes and what participants actually said. As analysis progressed layers of interpretation were applied but the origins of this interpretation were always grounded within the raw data facilitating a clear audit trail from data to interpretation.	Explanation of this point added under 'data analysis' Reference to 'themes emerging/arising' has been removed.	Pg. 7
	I think the sample/recruitment/eligibility criteria sections should be placed before analysis.	Thank you for your recommendation.	We have prepended Study Setting, Eligibility Criteria, Sample & Recruitment sections to Data Analysis.	Pg. 6, Pg. 7
	The protocol would also benefit from more detail on how the authors will ensure that data saturation is reached?	Thank you for also raising this point about data saturation. As outlined above we set a parameter for sample size based on previous studies and our aim for a diverse sample. We were also required to set a sample size for ethical opinion. Data saturation is a contentious issue within reflexive TA and this is now alluded to. While we are limited within this protocol paper to explore this issue in depth we will have more scope to examine if data saturation was reached, and what informed our conclusion on this, in the findings paper.	We have alluded to the current debate surrounding data saturation and provided further justification for the sample size determined at the outset of this study.	Pg. 6

VERSION 2 – REVIEW

REVIEWER	Shivanthi Balalla, PhD Auckland University of Technology, New Zealand.
REVIEW RETURNED	30-Dec-2020

GENERAL COMMENTS	Thank you for the amendments and revision of the manuscript. The revised version reads very clear and is very detailed in its steps. A few comments below. The authors might like to use the most current evidence from the Global Burden of Disease series to highlight the current burden of TBI. The following references might be of use: Haagsma JA, James SL, Castle CD, et al. Burden of injury along the development spectrum: associations between the Socio-demographic Index and disability-adjusted life year estimates from the Global Burden of Disease Study 2017. Injury Prevention 2020;26:i12-i26. One thing that still needs clarification, is the use of physicians involved in neurotrauma care - does this exclude surgeons? Surgeons perspectives' would be essential in this study. Line 53 (page 6) - you've introduced reflexive TA in abbreviated format - please write in full. Line 13 (page 7)- "local PIs" - if not defined in full earlier, please write in full i.e. "local primary investigators"
REVIEWER	Christel McMullan University of Birmingham
REVIEW RETURNED	05-Jan-2021
GENERAL COMMENTS	I have no further comments.